*A Nature Portfolio journal*

# Systems-level exploitation of OxyR regulon unravels a potential antibacterial target in *Pseudomonas aeruginosa*
Kaiyu Cui[1,2,11], Zhiyu Fu[3,11], Ruiling Sun[1,4], Qiuyun You[1], Fei Wang[5], Xuan Sun[6], Yong Tan[7], Yuwen Xia[7], Ping Wang [2] ✉, Qing Wei [8,9] ✉, Dao Wang [3] ✉ & Weifeng Yang [10] ✉

Bacteria coordinate the response to oxidative stress through LysR-type transcriptional regulator (LTTR) OxyR. However, only fragmentary information on the regulation and function of OxyR has been gleaned in the opportunistic pathogen *Pseudomonas aeruginosa*. Here, we delineate the OxyR regulon using multi-omics analyses. OxyR is found to positively regulate several genes involved in quorum sensing (QS) and energy metabolism. OxyR is further involved in the negative regulation of amino acid transporters that was confirmed by metabolomics analysis. Finally, we uncover *gltS*, an OxyR regulon gene, could be used as a potential drug potentiation target. Altogether, our results confirm that, apart from its dominant role in defense against oxidative stress in *P. aeruginosa*, OxyR acts as a global regulator of QS, energy metabolism and amino acid homeostasis, but also serves as a model system to identify potential antibacterial target such as *gltS*.

*Pseudomonas aeruginosa* is a ubiquitous, Gram-negative bacterium that inhabits diverse ecological niches such as soils, marshes, and coastal marine waters as well as the surfaces of plants and animals[1]. As an opportunistic pathogen, *P. aeruginosa* is able to infect plants, animals, and humans, and has been recognized as one of the primary causative agents of nosocomial diseases with generalized symptoms such as inflammation and sepsis[2–4]. Due to its versatile adaptability and diverse metabolism and ever-mounting evidence for increased antibiotic resistance, diseases caused by *P. aeruginosa* represent a significant health threat despite significant advances in treatment alternatives.

During infection, *P. aeruginosa* produces a battery of virulence factors that contribute to colonization and infectivity, among which are the phenazine pyocyanin, the fluorescent siderophore pyoverdine and serine proteases such as elastase[5–7]. Expression of a large subset of those virulence factor-encoding genes is controlled via an intercellular, cell-to-cell communication system commonly referred to as quorum sensing (QS), whereby *P. aeruginosa* produces, secretes and responds to small extracellular signaling molecules called autoinducers (AI)[8–10]. In addition, QS in *P.*

*aeruginosa* also regulates several other cellular processes including biofilm formation, adhesion, and genes involved in the oxidative stress response[11].

As a consequence of its growth in highly diverse environmental niches, *P. aeruginosa* has to adapt to ever-changing environments, one of which is when it is exposed to a variety of oxidants. The most prominent and commonly found oxidants are **r**eactive **o**xygen **s**pecies (ROS), including hydrogen peroxide ($H_2O_2$), superoxide anion ($O_2^-$) and hydroxyl radical (OH·)[12,13]. As one of the major challenges for living organisms, ROS cause damage to DNA, lipid membranes, proteins, and cofactors, and are therefore implicated in numerous human degenerative diseases (e.g., aging, atherosclerosis, inflammation, etc.). In addition, in contact with free ferrous iron, $H_2O_2$ is further reduced to the most damaging oxygen radical, OH·, via the Fenton reaction[14]. OH· reacts with virtually all known biomolecules at diffusion-limited rates.

Protection from oxidative stress is largely based on several defense mechanisms, including the production of antioxidant enzymes (catalase, superoxide dismutase, and peroxidase), iron sequestering proteins such as bacterioferritin and ferritin[15], proteins like thioredoxins, glutaredoxin as

¹School of Pharmaceutical Sciences, Hubei Shizhen Laboratory, Hubei University of Chinese Medicine, Wuhan, China. ²Engineering Research Center of TCM Protection Technology and New Product Development for the Elderly Brain Health, Ministry of Education, Wuhan, China. ³Department of Pediatrics, The First Affiliated Hospital of Zhengzhou University, Zhengzhou, China. ⁴Department of Clinical Laboratory, Wuhan Fourth Hospital, Wuhan, China. ⁵Department of Encephalopathy, Wuhan Hospital of Traditional Chinese Medicine, Wuhan, China. ⁶School of Basic Medical Sciences, Hubei University of Chinese Medicine, Wuhan, China. ⁷Institute of Basic Research in Clinical Medicine, China Academy of Chinese Medical Sciences, Beijing, China. ⁸Shanghai Cinopath Medical Laboratory Co., Kindstar Globalgene Technology Inc, Shanghai, China. ⁹Kindstar Global Precision Medicine Institute, Wuhan, China. ¹⁰Experimental Research Center, China Academy of Chinese Medical Sciences, Beijing, China. ¹¹These authors contributed equally: Kaiyu Cui, Zhiyu Fu. ✉e-mail: 1008@hbucm.edu.cn; weiqing@cinodx.com; deai315@163.com; sunzhuyang@126.com

**Fig. 1 | Comprehensive OxyR regulon in *P. aeruginosa*. A** Functional classification of *P. aeruginosa* OxyR regulon under hypoxia condition. Functional classes were obtained from www.pseudomonas.com[82]. Two groups of values (promoted genes and repressed genes in the *oxyR* mutant) were chosen and compared. Values indicate the number of *oxyR*-controlled genes within the respective classes. **B** qRT-PCR validation of the microarray data. Randomly selected genes from were checked with quantitative real-time PCR analysis as described in Materials and Methods. All data were obtained from at least two independent experiments with at least three replicates. **C** Mean log$_2$ ratios of the qRT-PCR data are plotted against the mean log2 ratios of the RNA-seq data.

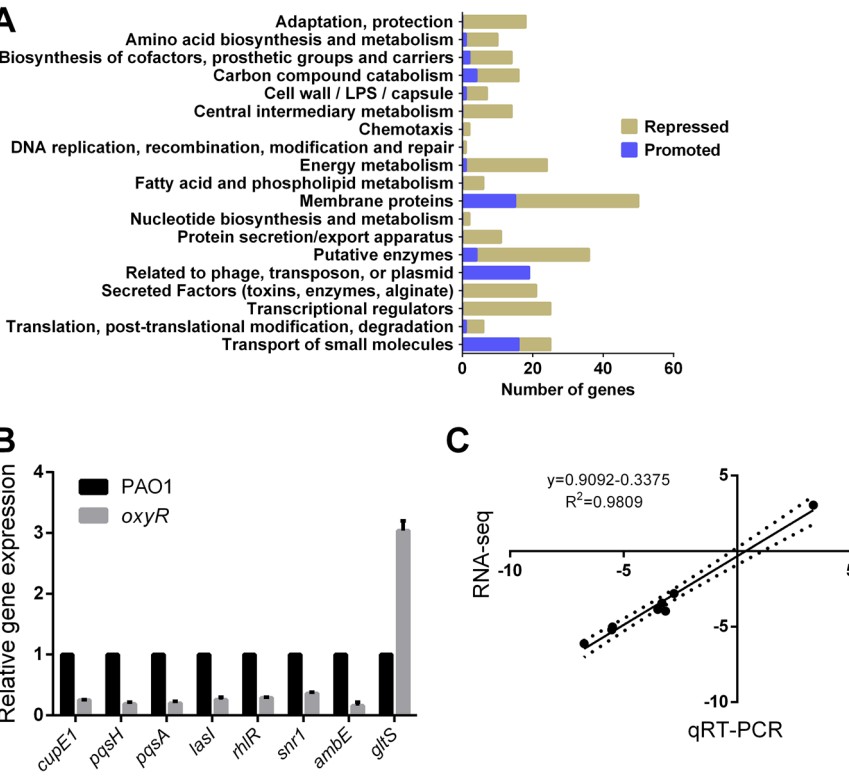

well as the tripeptide, glutathione[16]. Extensive studies have focused on *Escherichia coli* as a model organism to investigate the regulation of oxidative stress responses. Many of these studies revealed that the bacterial genetic responses to oxidative stress are controlled through two major transcriptional regulators, OxyR and SoxR[12,17–20]. $H_2O_2$ activates the LysR-type transcriptional regulator (LTTR) OxyR via the oxidation of two conserved cysteines (Cys 199 and Cys 208) and formation of an intramolecular disulfide bond[19,21]. In *E. coli*, the oxidized form of OxyR induces the expression of a set of defensive genes including *katG*, *ahpCF*, *dps*, *gorA*, *grxA*, and *oxyS*[20,22,23].

Similarly, to defend itself against oxidative stress, *P. aeruginosa* produces two superoxide dismutases (SOD, Mn-cofactored SodA and Fe-cofactored SodB), which represent the first line of defense against the superoxide anion $O_2^{·-}$ converting it to $H_2O_2$ while three catalases (KatA, KatB and KatE) protect the bacteria from $H_2O_2$. Finally, at least four alkylhydroperoxide reductases (AhpA, AhpB, AhpCF, and Ohr) detoxify $H_2O_2$ and several organic peroxides[24]. The OxyR homologue in *P. aeruginosa* is crucial for the upregulation of the antioxidant genes *katA*, *katB*, *ahpB*, and *ahpCF* after exposure to $H_2O_2$[24]. Other findings have revealed its role in the pathogenesis of *P. aeruginosa*[25], in the iron uptake mediated by the siderophore pyoverdine[26,27], and in the regulation of pyocyanin production[28] and type VI secretion system[29,30]. Aside from those, transcriptomic analyses have shown that when faced with $H_2O_2$, *P. aeruginosa* exhibits an upregulation of the protection mechanism and a downregulation of the primary metabolism, suggesting a link between both processes[31,32], possibly in part mediated by OxyR.

However, there is still little information concerning OxyR regulon in pathogenic bacteria including *P. aeruginosa* using RNA-seq analyses, especially under hypoxia. Recently, we described, via chromatin immunoprecipitation, that *P. aeruginosa* OxyR binds to other targets, suggesting that the OxyR regulon is not confined to the oxidative stress response genes[30]. In this study, we used transcriptomic and metabolomic analysis to identify the effects of *oxyR* disruption under normal growth condition (in the absence of exogenous $H_2O_2$) and determined that OxyR not only regulates the oxidative stress protection response, but also controls the expression of an array of QS-related genes and amino acid transporters. Further phenotypic

analysis has showed that several genes such as *gltS*, encoding a glutamate/sodium symporter, within OxyR regulon could be used to enhance the antibacterial killing, suggesting the potential role of *oxyR* in drug discovery. These findings shed light on the complexity of oxidative stress response in *P. aeruginosa* and provide valuable insights into the functionality of OxyR in bacteria.

## Results

### OxyR regulon revisited in *P. aeruginosa*

To gain insight into the regulatory breadth of OxyR in *P. aeruginosa*, we carried out a global analysis of the transcriptional responses without the addition of $H_2O_2$, to determine the number of OxyR-regulated genes under normal growth conditions. The reason to exclude the induction of extra oxidative stress via the addition of $H_2O_2$ lies in the possible secondary signals generated by the inducer, which distinguishes this study from previously published transcriptome analyses[22,31,32]. Our goal was, therefore, to determine the global effect of *oxyR* deletion under standard lab growth conditions for *P. aeruginosa*.

*P. aeruginosa* cells were harvested at the early stationary phase and RNA was extracted and processed according to the recommendations of Illumina system for RNA-seq analyses. After data qualification control and processing, we finally obtained a comprehensive data set summarized in Supplementary Table 1 and Fig. 1. Overall, our results exhibited high quality and accuracy since only little difference can be detected between two separate data sets by applying a stringent cut-off on *P*-value ($P < 0.01$, Supplementary Table 1). As shown in Fig. 1, when bacteria were grown in LB, disruption of *P. aeruginosa oxyR* caused changes in the expression of 510 genes (9% of all *P. aeruginosa* genes), among which 411 genes were repressed and 99 genes upregulated, suggesting that OxyR mainly acts as an activator. A complete list of differentially expressed genes is shown in Supplementary Table 1. Functional classification reveals that genes involved in adaptation, energy metabolism, membrane proteins, putative enzymes, secreted factors, and transcriptional regulators showed a decreased transcription level in *oxyR* mutant (Fig. 1A), while genes involved in the transport of small molecules as well as genes related to phage are largely increased at transcriptional levels (Fig. 1A). As anticipated, the typical OxyR

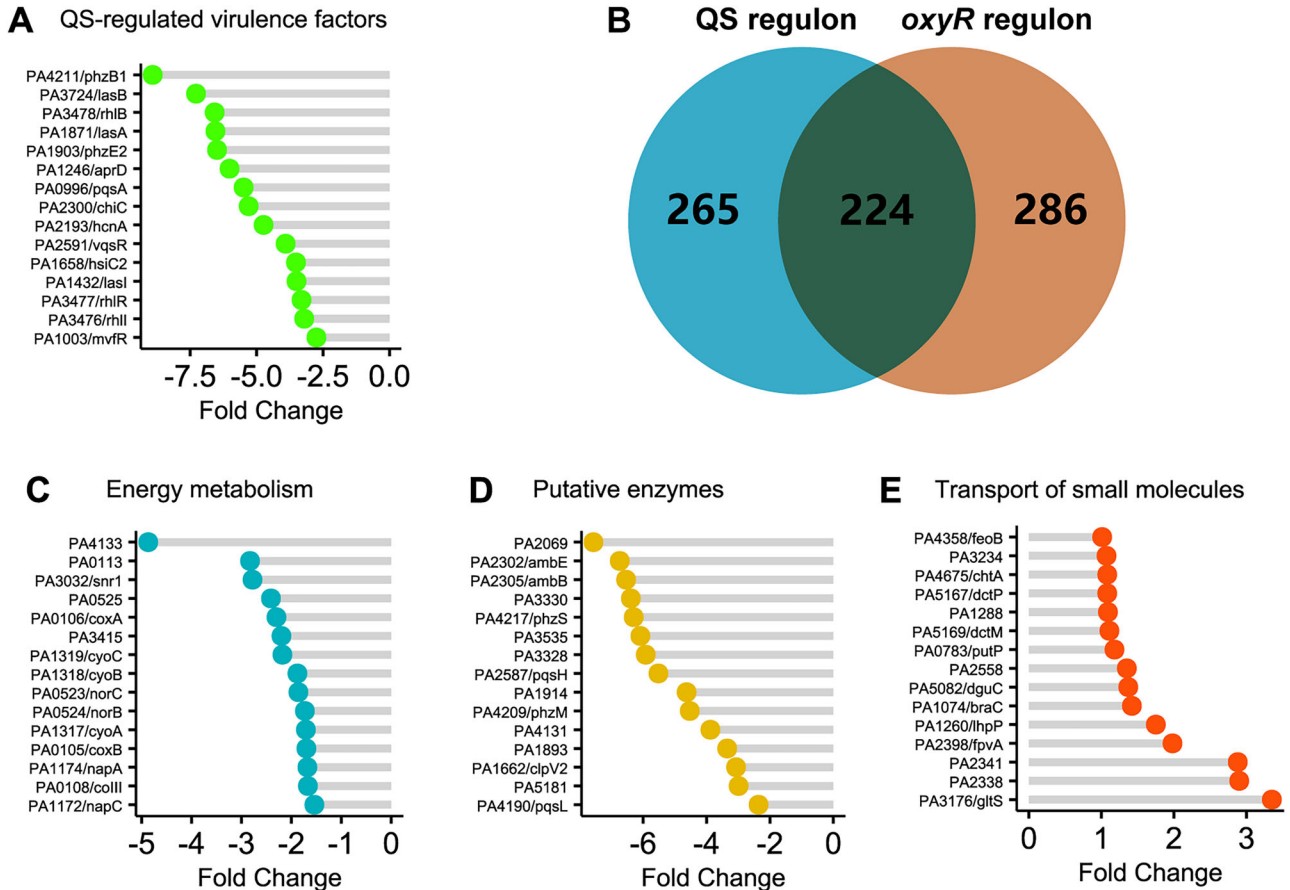

**Fig. 2 | Specific OxyR regulon in *P. aeruginosa*. A** OxyR positively regulates virulence associated genes. Top 15 genes involved in virulence are shown as representatives of QS-regulated genes. PA number and gene names (if available) are both shown. **B** OxyR regulon shared overlap with custom QS regulon. Venn diagrams showing the overlap between QS regulon and *P. aeruginosa* OxyR regulon.

More than 224 genes were shown in both regulons in *P. aeruginosa*. **C** OxyR positively regulates energy metabolism in *P. aeruginosa*. **D** OxyR positively regulates putative enzymes in *P. aeruginosa*, mainly of those enzymes belong to oxygenase. **E** OxyR negatively regulates amino acid homeostasis. Transporters of small molecules are selected to show.

regulon associated with the oxidative stress response, including *katE*, *snr1* and *pntAA* were found to be down-regulated in this study (Supplementary Table 1; −2.4, −2.8 and −1.1 in log2FC, respectively)[30], which clearly demonstrated *oxyR* mutation led to dysfunction of these genes.

In addition, a selection of differentially expressed genes from each condition was used to confirm the RNA-seq data using quantitative Real-Time PCR (qRT-PCR) and the results demonstrated good correlation with the RNA-seq results (linear correlation analysis, $R^2 = 0.9809$) (Fig. 1B, C). Furthermore, we have used Sanger sequencing to identify the integrity of *lasR* in our *oxyR* mutant and ruled out the possibility that *lasR* point mutations in the *oxyR* background. Since this mutation occurs in vitro and in vivo and might complicate the interpretation of our RNA-seq results.

**OxyR positively regulates virulence associated genes**
As can be seen from Supplementary Table 1 and Fig. 2A, the expression of several virulence genes was downregulated in the *oxyR* mutant, including QS-regulated genes. Specifically, two phenazine biosynthesis operons (*phzA1B1C1D1F1G1* and *phzA2B2C2D2F2G2*), amb antimetabolite biosynthesis operon (*ambABCDE*)[33], PQS signal synthase operon (*pqsABCDE*), hydrogen cyanide synthase operon (*hcnABC*), alkaline protease operon (*aprXDEFAI*), rhamnolipid synthesis operon (*rhlAB*), *lasA*, *lasB*, *chiC*, were highly repressed in *oxyR* mutant. In addition, a group of key regulators such as *rhlR*, *mvfR*, *vqsR*, *vqsM*, *pprA*, and *pprB* were also found to be repressed in *oxyR* mutant, explaining why these QS-regulated genes were downregulated and indicating that *oxyR* plays a vital role in the regulation of QS signaling in *P. aeruginosa*[28].

Given the inhibition of known QS systems in *P. aeruginosa*, we aimed to determine the overlap between OxyR regulon and QS regulon under typical growth conditions. We then integrated the previously identified QS regulon into a custom made QS regulon[34,35] and compared it with the revisited OxyR regulon. Clearly, we found that our determined OxyR regulon showed a close relationship with custom QS regulon (Fig. 2B). More than 200 genes (224) were identified to overlap with the custom QS regulon (489 genes) in *P. aeruginosa*, suggesting that *oxyR* has a substantial impact on the regulation of QS systems and clearly this positive regulation of QS systems will contribute to the complex regulatory network of *P. aeruginosa*.

In addition, genes involved in protein secretion were also down-regulated in the *oxyR* mutant. For instance, type VI secretion system (T6SS) gene clusters H2-T6SS (PA1656 to PA1669) and H3-T6SS (PA2360 to PA2373) were both repressed in the *oxyR* mutant, probably due to the inhibition of QS systems that positively regulate H2- and H3-T6SS expression[36,37]. Besides, the general secretion pathway (type II secretion system) was also repressed in the *oxyR* mutant, suggesting that the secretion of most signal-peptide-dependent exoproteins was probably decreased.

Altogether, we have unraveled that the expression of a large battery of genes involved in virulence is downregulated in *oxyR* mutant, indicating that OxyR acts as a positive regulator of virulence in *P. aeruginosa*.

**OxyR positively regulates energy metabolism in *P. aeruginosa***
It was previously reported that $H_2O_2$ treatment of *P. aeruginosa* led to decreased expression of primary metabolism, including genes involved in energy generation, respiration chain complex, and oxidative phosphorylation[31].

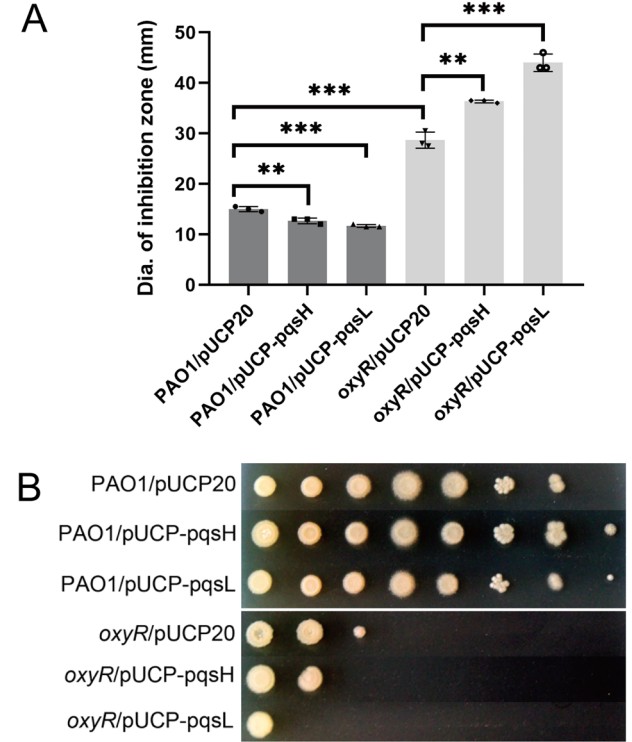

**Fig. 3 | Overproduction of PqsH and PqsL sensitizes the *oxyR* mutant to oxidative stress. A** Growth inhibition by $H_2O_2$ as determined by agar diffusion assays. Monooxygenase-producing strains (with pUCP-pqsH and pUCP-pqsL) in PAO1 exhibited an increased resistance towards $H_2O_2$ as compared to that of wild type (student's *t* test; *p* < 0.01; ∗, *p* < 0.001). However, monooxygenase-producing strains in the *oxyR* mutant exhibited an increased sensitivity towards $H_2O_2$ as compared to the control (student's *t* test; *p* < 0.01; ∗, *p* < 0.001; *n* = 3). The diameter of the inhibition zone is given as the mean ± SEM of triplicate samples. **B** Serial dilution analysis of the monooxygenase-producing strains as compared to the control. The results validated the growth inhibition effect of the monooxygenases in the *oxyR* mutant and the increased fitness when expressed in PAO1 wild type. The overnight bacterial cultures were diluted from $10^9$ to $10^2$ per ml.

In our study, we observed a tuning of energy metabolism in the *oxyR* mutant (Fig. 2C and Supplementary Table 2), suggesting that without protection from intact OxyR, *P. aeruginosa* cells are undergoing oxidative stress similar to $H_2O_2$ exposure. Among these genes, the heme-copper oxidase superfamily of enzymes, cytochrome *c* and ubiquinol oxidases were highly repressed at the transcriptional level. The expression of genes involved in denitrification was also decreased in the *oxyR* mutant compared with wild type PAO1, suggesting that the *oxyR* mutant was undergoing reduced cellular respiration and denitrification even without the addition of extra oxidants.

In addition, we noticed that in the *oxyR* mutant, the expression of multiple monooxygenases was reduced (Fig. 2D and Supplementary Table 2). As we mentioned earlier, we observed a downregulation of PQS biosynthesis genes (*pqsABCDE* operon) in the *oxyR* mutant (Supplementary Table 1). Interestingly, the *pqsH* gene was found to be downregulated (46-fold) when the *oxyR* mutant was grown in LB. This is of note because PqsH catalyzes the terminal reaction from 2-heptyl-4-quinolone (HHQ) to PQS, a process involving oxygen ($O_2$) and NADH[38]. It is worth to mention that PQS was also demonstrated to exhibit both oxidant and antioxidant properties[39] partly because of its iron-chelating properties[40,41]. When we overproduced *pqsH* in the *oxyR* mutant using a constitutive expression system, we found that increased PqsH production sensitized the *oxyR* mutant to oxidative stress and led to a decreased survival rate in a serial dilution analysis in *oxyR* (Fig. 3A and 3B). Likewise, another FAD-dependent monooxygenase, PqsL, involved in HQNO production[42], when overproduced in the *oxyR* mutant renders

cells even more sensitive to oxidative stress (Fig. 3A and 3B). In contrast, overexpression of *pqsH* and *pqsL* in wild-type PAO1 cells elicited the opposite response, namely, a higher plating efficiency in LB (Fig. 3B), which could be explained by the induction via the functional OxyR of the expression of a battery of defensive genes.

To resist the challenge of iron-related Fenton reaction, we have noticed that in the *oxyR* mutant, several iron-associated genes, such as *fpvF*, *fpvG*, and *feoB* (Supplementary Table 1) were upregulated, indicating that OxyR negatively regulates these genes and related processes.

Altogether, we have uncovered the tuning of a repertoire of energy metabolism-associated genes as one of the underlying protective mechanisms in *P. aeruginosa*.

### OxyR negatively regulates amino acid homeostasis

To further understand the survival strategies adopted by *oxyR*, we noticed that a wide variety of genes encoding transporters of small molecules were altered, especially these involved in amino acid transport (Fig. 2E and Supplementary Table 3). Homeostasis of more than 13 amino acids were affected in the *oxyR* mutant, indicating that OxyR mediated this newly identified process in bacteria to resist against oxidative stress. Among these genes, *gltS* (PA5176), encoding a transporter involved in glutamate/sodium symport, showed the highest expression level in the *oxyR* mutant (10-fold increase). In addition, genes encoding transporters involved in branched-chain amino acids (leucine, isoleucine, and valine), aromatic amino acids (tryptophan, tyrosine, phenylalanine), lysine, arginine, ornithine, histidine, glycine-glutamate, proline, and alanine were all upregulated in *oxyR* mutant compared to wild type PAO1. This finding has raised the possibility whether the transport of these amino acids could be involved in oxidative stress defense.

To prove this hypothesis, we used untargeted metabolomics to verify these physiological changes in *oxyR* mutant compared to wild type PAO1. As shown in Fig. 4A and Supplementary Table 4, we showed that a wide array of pathways has been affected, especially the amino acid metabolism including cysteine and methionine, tryptophan, and glutamate. Furthermore, we have classified these differential metabolites into five major groups (Fig. 4B) and found that majority of these metabolites have been overproduced in *oxyR* mutant compared to wild type PAO1, indicating these metabolites are playing protection roles against ROS. Interestingly, most of them are amino acids or derivatives of amino acids, further suggesting the involvement of amino acids in defense against oxidative stress. In addition, one of the most prominent findings in our metabolite analysis is that we confirmed that *oxyR* mutant, which has cell death caused by ROS attack, underwent oxidation of guanine[43] and finally a reduced dGTP pool in cells (Fig. 4B).

To prove the potential involvement of amino acids in defense against oxidative stress, we sought to determine whether certain amino acid found in this study could restore or promote the growth of *oxyR* mutant under normal growth condition (Fig. 4C). We incorporated one of the amino acids, methionine, into LB agar plate and revealed that it could improve the growth of *oxyR* mutant under normal growth condition, where oxidative stress exists. Supplementation of cysteine and aspartate to LB agar plates gave rise to the same restoration results for *oxyR* mutant (Fig. 4C).

Altogether, we have determined that *oxyR* has been involved in amino acid homeostasis in bacterial cells.

### Elucidation of *gltS* as a drug potentiation target in *P. aeruginosa*

The growth defect of *oxyR* mutant has been described previously and it was noticed that under hypoxia, the *oxyR* mutant restored its growth as compared to wild type[28]. Furthermore, we have observed that under hypoxia, the transcription of *gltS* was greatly upregulated in *oxyR* mutant, prompting us to hypothesize that *gltS* might be involved in defense against oxidative stress in *P. aeruginosa*. To prove this, we constructed an in-frame deletion mutant of *gltS* and measured intracellular ROS production of *gltS* compared to PAO1 by using the redox dye dihydro-dichloro-fluorescein diacetate ($H_2DCFDA$). As shown in Fig. 5A, disruption of *gltS* could significantly

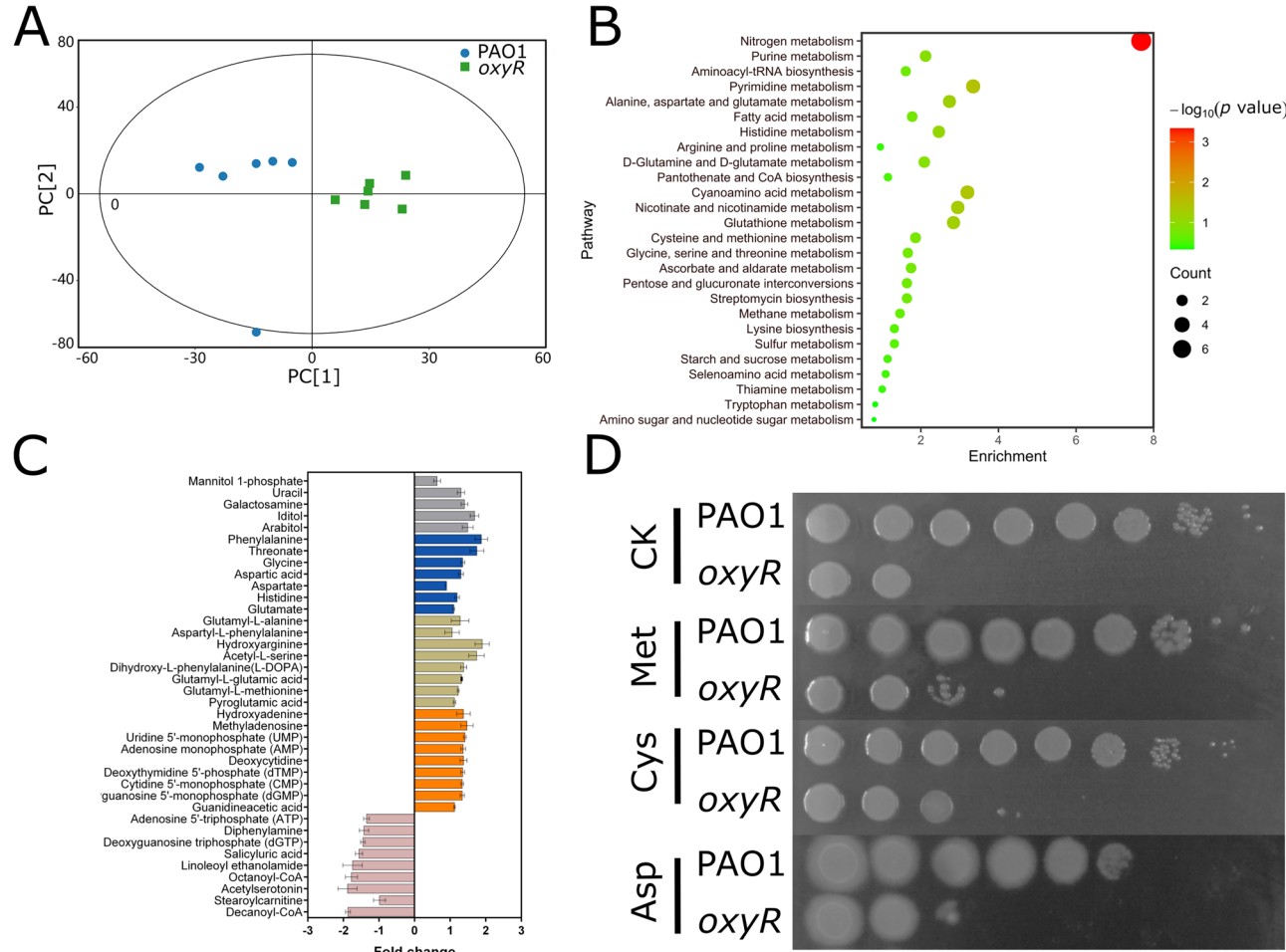

**Fig. 4 | Metabolomics analysis of *oxyR* in *P. aeruginosa*. A** Score plot of OPLS-DA (orthogonal projections to latent structures-discriminant analysis) model obtained from PAO1 and *oxyR* metabolic profiling data ($R^2X = 0.491$, $R^2Y = 0.885$, $Q^2 = 0.549$). Both groups showed significant separation and most data sets are located in Hotelling's T-squared ellipse ( > 95%), with one exception in PAO1 data set. (**B**) Pathway enrichment analysis of *oxyR* metabolites as compared to PAO1. **C** Differential metabolites in *oxyR* mutant. Gray bars, solvents; blue bars, amino acids; Khaki bars, amino acid derivatives; orange bars, nucleotides; pink bars, groups of reduced metabolites. Data are means ± SEM (n = 6). **D** Spot assay of amino acids in growth complement of *oxyR* mutant under aerobic condition. 5 µM methionine (Met), cysteine (Cys), and aspartate (Asp) were used in a final concentration supplemented in LB agar plates (CK). The overnight bacterial cultures were diluted from $10^9$ to $10^2$ per ml.

increase the production of ROS and complementing *gltS* in a low-copy shuttle vector pUCP20 could restore the ROS levels in *gltS* mutant to the wild type level. Our results clearly demonstrate that GltS is involved in the generation of endogenous ROS in *P. aeruginosa*.

It is widely acknowledged that bactericidal antibiotics share a common mechanism of action through generation of ROS to mediate cell death[44]. Furthermore, the generation of endogenous ROS in bacterial cells could potentiate the action of bactericidal antibiotics[45]. To test whether *gltS* could be used as a drug potentiation target, we determined bacterial killing during treatment with the aminoglycoside gentamicin and tobramycin, the β-lactam ampicillin and the fluoroquinolone ciprofloxacin. We found that the *gltS* mutant presented an increased sensitivity to the aminoglycosides (Fig. 5B, C) and β-lactam (Fig. 5D).

To further explore the mechanisms underlying *gltS*-mediated drug potentiation, we initially conducted growth curve analyses comparing the *gltS* mutant, its parental strain PAO1 and complementary derivates. As depicted in Supplementary Fig. 1A, the *gltS* mutant exhibited robust growth fitness comparable to PAO1, while the complementary strain with functional *gltS* showed intermediate growth. Notably, overexpression of *gltS* resulted in growth inhibition. These findings effectively dispelled the notion that the impaired survival of *gltS* in the presence of antibiotics stemmed from slower growth. Subsequently, we monitored the growth of these strains over 24 h under various antibiotic treatments. It was striking that treatment with bactericidal antibiotics gentamicin (Supplementary Fig. 1B), tobramycin (Supplementary Fig. 1C), and ampicillin (Supplementary Fig. 1D), significantly impaired the growth of *gltS* mutant relative to PAO1, solidifying the link between *gltS* disruption and heightening antibiotic susceptibility in *P. aeruginosa*. In contrast, the *gltS* mutant displayed no growth inhibition when exposed to the bacteriostatic antibiotics tetracycline (Supplementary Fig. 1E), underscoring the specificity of the observed effects.

We next asked whether GltS is conserved among ESKAPE pathogens (*Enterococcus faecium*, *Staphylococcus aureus*, *Klebsiella pneumoniae*, *Acinetobacter baumannii*, *P. aeruginosa*, and *Enterobacter spp.*) and therefore could be broadly used as a drug potentiation target to treat bacterial infections[46]. As can be seen from Fig. 5E, *gltS* is presents in all ESKAPE pathogens except in *Enterococcus faecium*, suggesting that GltS is well conserved among pathogens (5 out of 6 major pathogens, 83.3% prevalence).

In summary, we have elucidated that *gltS* is involved in endogenous ROS production and could be used as a potential drug potentiation target to enhance antibacterial killing by bactericidal antibiotics, especially aminoglycosides and β-lactams.

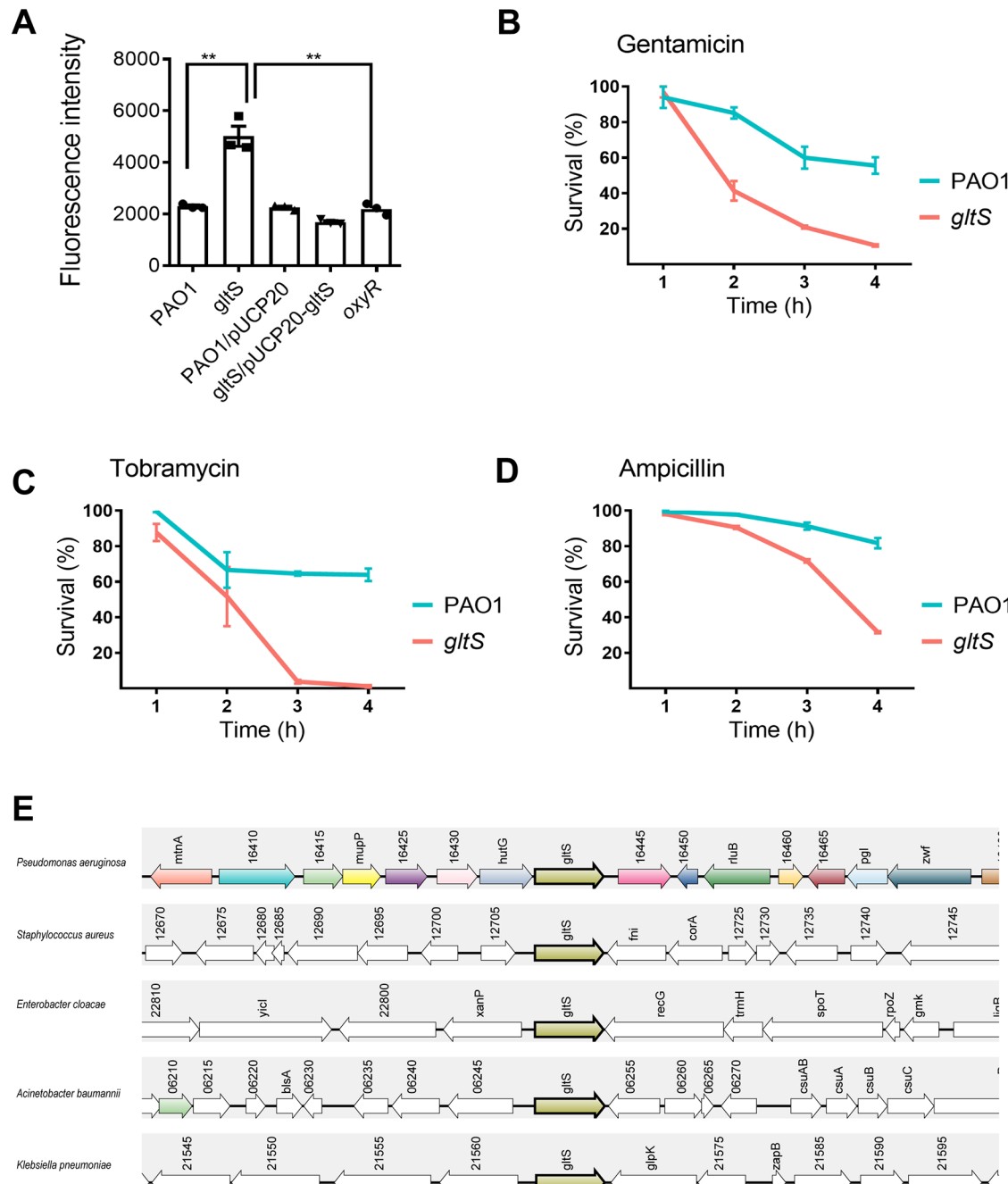

**Fig. 5 | Elucidation of *gltS* as a drug potentiation target in *P. aeruginosa*. A** ROS generation in *gltS*, PAO1, and its derivatives. The median of fluorescence intensity of 50,000 log phase grown cells treated with 20 mM H₂DCFDA is given. The assays were performed three times with independent cultures. Data are means ± SEM. Significance was determined by Student's *t* test; $p < 0.01$; $n = 3$. Evaluation of susceptibility to killing by antibiotics. Time course survival analysis of *gltS* and PAO1 treated with gentamicin (**B**), tobramycin (**C**) and ampicillin (**D**). Error bars are indicated, mean ± SEM for all plots. Student's *t* test; $p < 0.01$; n = 3. (**E**) Synteny analysis of GltS in ESKAPE pathogens. Complex syntenies were obtained by using *P. aeruginosa* GltS protein sequence as query sequence to search against indicated Pseudomonads species. A consistent color coding permitted the correct identification of both orthologs and paralogs. The genes corresponding to the query proteins are drawn in bold and boxed with light olive color.

## Discussion

OxyR is an important regulator of the oxidative stress response in bacteria that functions as a H₂O₂ sensor to activate gene expression via the formation of an intermolecular disulfide bond[19]. In the model organism *E.coli*, it was reported that OxyR could not only control the expression of the oxidative stress response genes but is also implicated in other processes such as iron uptake, Fe-S cluster assembly and repair, heme biosynthesis and sRNA regulation[22,23,47]. In the human opportunistic pathogen *P. aeruginosa*, it is acknowledged that OxyR is involved in the oxidative stress response genes,

iron uptake, virulence, and the production of the phenazine compound pyocyanin[25–28]. In addition, transcriptomic analyses have shown that exposure to H₂O₂ in *P. aeruginosa* resulted in an upregulation of genes involved in protection mechanisms and a downregulation of those for primary metabolism[31,32]. More recently, structural details of OxyR in diverse bacteria provided valuable insights into the mode of action of peroxide sensing[48–50].

In this report, we demonstrate that *P. aeruginosa* OxyR is engaged in more processes than previously appreciated, particularly via its connection

with QS, energy metabolism, and transport of small molecules (Fig. 2). By using systems-level transcriptomic and metabolic analysis, we revealed (i) the mechanism of survival of *oxyR* in hypoxia, involving the downregulation of energy metabolism, virulence, and putative enzymes, particularly monooxygenases, such as *pqsL* and *pqsH* (Fig. 3); (ii) the OxyR regulon in routine growth conditions confirms the strong link with QS, suggesting that *oxyR* is a master regulator of virulence; (iii) the OxyR regulon provides insights into the antibacterial killing by antibiotics, providing substantial number of novel drug targets in bacteria. Therefore, we have not only uncovered the OxyR regulon in *P. aeruginosa*, but also established its potential application in antibiotics discovery against human pathogens.

To survive in aerobic environments, the *oxyR* mutant has evolved numerous strategies to combat the attack of ROS. One of the known strategies is the production of protective pigments such as pyocyanin to complement the growth defect of the *oxyR* mutant in aerobic conditions[28]. This is due to the reversible redox trait of pyocyanin[51]. In this study, we have unraveled another survival strategy adopted by *oxyR*, which involves the downregulation of genes controlled by QS, genes encoding putative enzymes such as monooxygenases, and genes involved in energy metabolism. For example, we have confirmed that overexpression of monooxygenases *pqsL* and *pqsH* sensitized *P. aeruginosa* to oxidative stress (Fig. 3), indicating that endogenous utilization of oxygen and subsequent production of ROS are simultaneously occurring in bacterial cells. In line with this result, Schertzer and colleagues confirmed that PqsH catalyzes the terminal step in PQS production using the substrates HHQ, NADH and oxygen[38]. Furthermore, PQS has also been shown to have pro-oxidant and iron-chelating activity, both of which are related to oxidative stress response[39,40]. Therefore, *P. aeruginosa* has evolved strategies to avoid the use of oxygen and generation of ROS to survive in aerobic conditions. Besides this strategy, downregulation of energy metabolism such as cytochrome c-mediated respiration (Fig. 2) has been further employed to mitigate the effect of oxidative stress, consistent with previous findings[31,52].

Interestingly, we discovered a large repertoire of transporters of small molecules that have been upregulated by our systems-level transcriptomic and metabolic analysis of *oxyR* mutant compared to wild type PAO1. Our conclusion has been reinforced by a study of *Acinetobacter baumannii* exposed to $H_2O_2$[53]. Metabolic analysis has provided complementary measurement of amino acid influx in bacterial cells and offered opportunities to identify new metabolites or biomarkers for certain biological processes. In our study, this combinatory method has led to the identification of amino acids such as methionine, cysteine and aspartate necessary for the recovery of growth defect of *oxyR* mutant under aerobic condition (Fig. 4). Sulfur-containing amino acids such as methionine and cysteine are the most easily oxidized and these oxidation events are usually reversible and physiologically relevant[54–56]. In *E. coli*, it was found that proline metabolism could increase oxidative stress resistance[57]. Also, it was confirmed that glutamate could promote survival of *Neisseria meningitidis* during evasion of polymorphonuclear neutrophil leukocytes[58]. Therefore, bacterial amino acid homeostasis has arisen as a survival tactic to cope with oxidative stress.

The unexpected result from our study is that we have leveraged the OxyR regulon for the identification of one potential drug potentiation target, *gltS*. We have shown that deletion of *gltS* led to enhanced sensitivity to aminoglycosides and β-lactam, which was probably caused by the accumulation of ROS in *P. aeruginosa* cells. Previously, the potentiation of endogenous microbial ROS production has been utilized to enhance the antibacterial activity by oxidants and antibiotics[45]. Our study further strengthened this finding by extending endogenous ROS-potentiating gene repertoire to include amino acid transporters into ROS metabolic pathways[45]. Further study found that clinically relevant mutations in core metabolic genes confer antibiotic resistance and it was particularly interesting to note that glutamate synthase encoding genes *gltB*, *gltD*, and *gltA* were found to acquire mutations during antibiotic treatment[59]. Both studies strongly support our finding and conclusion that *gltS* could be potentially useful in antibiotic potentiation treatment. However, the idea that ROS mediated bacterial cell death has faced significant challenges, as several technical and biological concerns have emerged[60–62]. Therefore, it would be essential to conduct rigorous experiments to exclude the potential involvement of factors other than ROS in *gltS*-mediated antibiotic potentiation.

The predicted function of *gltS* is associated with glutamate/sodium ion symport, yet it has been seldom studied[63–65]. Previous study proved that disruption of *gltS* did not affect the utilization of glutamate but did impair the utilization of N-acetyl-L-glutamate[65]. It was further hypothesized that *gltS*, acting as a novel transporter, collaborates with its upstream gene (PA3175, a homologue of formiminoglutamate hydrolase) to form an operon for N-acetyl-L-glutamate utilization. In our study, we generated an in-frame deletion mutant of *gltS* and observed robust growth compared to wild type PAO1. Conversely, overexpression of *gltS* resulted in growth inhibition. The distinct growth response of the *gltS* mutant to bactericidal and bacteriostatic antibiotics suggest a novel role for *gltS*, potentially linking N-acetyl-L-glutamate utilization to the ROS response. The detailed mechanisms involved in these processes need further investigation.

Altogether, we have revisited the OxyR regulon and found OxyR acts as a global regulator of QS, energy metabolism and amino acid homeostasis. Importantly, by using multi-omics techniques and phenotypic verification, we have confirmed our results and discovered that *gltS*, one of the OxyR regulon genes, could be used as a potential drug potentiation target in pathogenic bacteria, such as *P. aeruginosa*. Our future work will focus on searching for more potential drug targets by using *oxyR* as a model and for drugs specifically targeting *gltS* as well as the molecular mechanisms underlying *gltS*-mediated antibiotic potentiation.

## Methods
### Bacterial strains and culture conditions
The *P. aeruginosa* wild-type PAO1 strain, its isogenic mutant *oxyR*, *gltS* as well as other derivative strains were used in this study and triplicate cultures were grown in Luria-Bertani (LB, BD) at 37 °C. When necessary, various class of antibiotics and other chemicals, such as glutamine, were supplemented at the specified concentrations.

### Spot assay
0.1 M stock solutions of L-Asp, L-Met, L-Cys, and L-His (Biotopped Technology Co., Ltd., Beijing) were prepared. The solutions were sterilized using a 0.22 μm filter to ensure aseptic conditions and stored at 4 °C for subsequent use. PAO1 and its derivatives were selected and cultured in 5 mL of LB liquid medium. A total of 28.5 mL of LB agar medium was aliquoted into square petri dishes pre-marked with scale lines. To do this, 1.5 mL of each amino acid solution was added to achieve a final concentration of 5 mM for each amino acid. The medium was gently agitated to ensure homogenous distribution of the amino acids. The overnight bacterial cultures were serially diluted to a range of concentrations from $10^{-1}$ to $10^{-9}$. Aliquots of 5 μL from each dilution were spotted onto the prepared LB agar plates containing the amino acids. The inoculated plates were incubated in a 37 °C constant temperature incubator. The growth of the bacterial colonies was monitored and recorded over a set period.

### Growth curve
Overnight cultures of bacterial strains in LB were diluted (1:100) in 3 ml LB medium and precultures incubated aerobically at 37 °C in a shaker at 200 rpm to an OD600nm of 0.5. The precultures were further diluted (1:100) in 1 ml LB medium. Growth was then analyzed in 96-wells microtitre plate containing 100 μl LB medium to which 100 μl of diluted precultures containing $10^5$ cells was added to obtain a final 1:200 dilution. The microtitre plates were incubated for 24 h at 37 °C in SpectraMax ID3 (Molecular Devices, USA) using the following settings: absorbance measured every 1 h at 600 nm after shaking for 10 s. Each culture was prepared in triplicate.

### Molecular manipulation
Isolation of plasmid and genomic DNA from *E. coli* and restriction enzyme digestion were performed according to the manufacturer's instructions

(Qiagen). DNA cloning, transformations and agarose gel electrophoresis were done as previously described[66]. In-frame and unmarked gltS deletion mutant in *P. aeruginosa* PAO1 was constructed using the suicide vector pEX18Ap as described with modifications[67,68]. Briefly, approximately 500 bp upstream and downstream sequences flanking *gltS* were amplified by PCR and fused with overlapping PCR. The PCR product was excised from the agarose gel and cloned into the pEX18Ap vector, resulting in the plasmid pEX18Ap-*gltS*. The pEX18Ap-based deletion allele was mobilized to *P. aeruginosa* and integrated into the chromosome by single crossover using *E. coli* S17-1 λpir as the delivery strain. Double crossover events were subsequently selected by growth in the presence of 5% sucrose. The *gltS* deletion mutant was confirmed by both PCR and sanger sequencing. Complementation analysis was done by cloning the intact *gltS* gene into the shuttle vector pUCP20 to generate the pUCP-gltS and introducing into the deletion mutants and wild type PAO1. The empty vector pUCP20 was also introduced to the *gltS* mutant and PAO1 as a control.

## RNA extraction

RNA extraction and subsequent sequencing data processing were carried out according to our previous study[69], with minor modifications. Overnight cultures of *P. aeruginosa* in LB were used to inoculate fresh LB medium in a 1:1000 dilution. After 12 h of incubation at 120 rpm at 37 °C, one ml of culture with an OD600nm of ~2.0 was immediately fixed with 2 ml of RNA Protect Reagent (Qiagen), following the manufacturer's instructions, and the fixed cell pellets were frozen at −80 °C until further use. All experiments were performed with three technical replicates. Total RNA was extracted using TRIzol® Reagent according to the manufacturer's instructions (Invitrogen) and genomic DNA was removed using RNase-free DNase I (TaKaRa). Then RNA quality was determined using 2100 Bioanalyzer (Agilent) and quantified using the ND-2000 (NanoDrop Technologies). High-quality RNA sample (OD260/280 = 1.8 ~ 2.2,  OD260/230 ≥ 2.0,  RIN ≥ 6.5,  28S:18S ≥ 1.0, >10 μg) was used to construct the sequencing library.

## Library preparation, and Illumina Hiseq sequencing

RNA-seq strand-specific libraries were prepared following TruSeq RNA sample preparation Kit from Illumina (San Diego, CA), using 5 μg of total RNA. Briefly, rRNA was removed by RiboZero rRNA removal kit (Epicenter), fragmented using fragmentation buffer. cDNA synthesis, end repair, A-base addition and ligation of the Illumina-indexed adaptors were performed according to Illumina's protocol. Libraries were then size selected for cDNA target fragments of 200 ~ 300 bp on 2% Low Range Ultra Agarose followed by PCR amplified using Phusion DNA polymerase (NEB) for 15 PCR cycles. After quantification by TBS380 Mini-Fluorometer, paired-end libraries were sequenced by Shanghai Biozeron Biotechnology Co.,Ltd (Shanghai, China) with the Illumina HiSeq PE 2 × 151 bp read length.

## Reads quality control and mapping

The raw paired end reads were trimmed, and quality controlled by Trimmomatic with default parameters[70]. Then clean reads were separately aligned to the reference genome (*Pseudomonas aeruginosa* PAO1, Accession number NC_002516) with orientation mode using Rockhopper software[71,72], which was a comprehensive and user-friendly system for computational analysis of bacterial RNA-seq data. As input, Rockhopper takes RNA sequencing reads generated by high-throughput sequencing technology to calculate gene expression levels with default parameters.

## Differential expression analysis and functional enrichment

To identify DEGs (differential expression genes) between two different samples, the expression level for each transcript was calculated using the fragments per kilobase of read per million mapped reads (RPKM) method. The method edgeR was used for differential expression analysis[73]. The DEGs between two samples were selected using the following criteria: the logarithmic of fold change was greater than 2 and the false discovery rate (FDR) should be less than 0.05. To understand the functions of these differential

expressed genes, GO functional enrichment and KEGG pathway analysis were carried out by Goatools[74] and KOBAS[75], respectively. DEGs were significantly enriched in GO terms and metabolic pathways when their Bonferroni-corrected *P*-value was less than 0.05.

## Quantitative real-time PCR (qRT-PCR)

Bacterial cells were harvested in the stationary phase, bacterial RNA was extracted by using RNeasy Midi Kit (Qiagen). The purity and concentration of the RNA was determined by NanoDrop ND-1000 spectrophotometer (Thermo Scientific). First-strand cDNA was reverse transcribed from one microgram of total RNA by using First-strand cDNA Synthesis Kit (Amersham Biosciences). qRT-PCR was performed in a Bio-Rad iCycler with Bio-Rad iQ SYBR Green Supermix (Bio-Rad). For all primer sets, the following cycling parameters were used: 94 °C for 3 min, followed by 40 cycles of 94 °C for 30 s, 55 °C for 45 s, and 72 °C for 30 s, followed by 72 °C for 7 min. The outer membrane lipoprotein *oprI* gene was used to normalize gene expression[76]. Amplification products were verified by electrophoresis on a 0.8% agarose gel. For statistical analysis of relative gene expression, the $2^{-\Delta\Delta CT}$ method was used[77]. All experiments were carried out in triplicate with three biological replicates. Signification difference was considered when *p* value < 0.5 using Student's *t* test.

## Metabolites extraction

Metabolomics analyses were carried out according to previous studies[78,79], with minor modifications. 60 mg of sample was taken and placed in an EP tube, then add 1200 μL extraction liquid (V methanol: V acetonitrile: V water = 2:2:1, which was kept at −20 °C before extraction) and 20 μL internal standard. Homogenized in a ball mill for 4 min at 45 Hz, then ultrasound treated for 5 min (incubated in ice water). After homogenization for 3 times, incubation for 1 h at −20 °C to precipitate proteins. Then centrifuged at 12000 rpm for 15 min at 4 °C. Transfer the supernatant (320 μL) fresh into EP tubes, dry the extracts in a vacuum concentrator without heating, add 100 μL extraction liquid (V acetonitrile: V water = 1:1) reconstitution. Vortex 30 s and sonicate 10 min (4 °C water bath), centrifuge for 15 min at 12000 rpm, 4 °C. Transfer the supernatant (60 μL) into a fresh 2 mL LC/MS glass vial, take 10 μL from each sample and pooled as QC samples, Take 60 μL supernatant for the UHPLC-QTOF-MS analysis.

## LC-MS/MS analysis

LC-MS/MS analyses were performed using an UHPLC system (1290, Agilent Technologies) with a UPLC BEH Amide column (1.7 μm, 2.1*100 mm, Waters) coupled to TripleTOF 6600 (Q-TOF, AB Sciex). The mobile phase consisting of 25 mM $NH_4OAc$ and 25 mM $NH_4OH$ in water (pH = 9.75) (A) and acetonitrile (B) was carried out with elution gradient as follows: 0 min, 95% B; 7 min, 65% B; 9 min, 40% B; 9.1 min, 95% B; 12 min, 95% B, which was delivered at 0.5 mL min⁻¹. The injection volume was 2 μL. The Triple TOF mass spectrometer was used for its ability to acquire MS/MS spectra on an information-dependent basis (IDA) during an LC/MS experiment. In this mode, the acquisition software (Analyst TF 1.7, AB Sciex) continuously evaluates the full scan survey MS data as it collects and triggers the acquisition of MS/MS spectra depending on the preselected criteria. In each cycle, 12 precursor ions whose intensity greater than 100 were chosen for fragmentation at a collision energy (CE) of 30 V (15 MS/MS events with a product ion accumulation time of 50 msec each). ESI source conditions were set as following: Ion source gas 1 as 60 Psi, Ion source gas 2 as 60 Psi, Curtain gas as 35 Psi, source temperature 650°C, Ion Spray Voltage Floating (ISVF) 5000 V or 4000 V in positive or negative modes, respectively.

## Data preprocessing and annotation

MS raw data (.d) files were converted to the mzXML format using ProteoWizard and processed by R package XCMS (version 3.2). The preprocessing results generated a data matrix that consisted of the retention time (RT), mass to charge ratio (m/z) values, and peak intensity. R package

CAMERA was used for peak annotation after XCMS data processing[80,81]. In-house MS2 database was applied in metabolite identification.

## Statistical and reproducibility

The data of virulence factor production, transcriptional analysis, and growth curve test were analysed by one-way ANOVA. Student's *t* test was used when one-way ANOVA showed significant differences ($P < 0.05$). All statistical analyses were performed with GraphPad Prism statistical software (GraphPad Software, La Jolla, USA) with the assistance of Excel (Microsoft).

## Reporting summary

Further information on research design is available in the Nature Portfolio Reporting Summary linked to this article.

## Data availability

The data that support the findings of the current study are available online or upon reasonable request. The RNA-seq datasets have been deposited in National Center for Biotechnology Information (NCBI) with an accession number GSE156842. The source data underlying the graphs and charts in the figures are uploaded as Supplementary Data. The values underlying Figs. 1A-C, 2A-D, 3A, 4C, 5A-D, and Supplementary Fig. 1A-E can be found in Supplementary Data.

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

## Acknowledgements

This work was supported by The Fundamental Research Funds for the Central Public Welfare Research Institutes (N0.JJPY2022014), Training Object of the Second Medical Leading Talent Project in Hubei Province, the Hubei Famous Doctor Studio Project of the Health Commission of Hubei Province (e.w.t. [2019] No. 47), the Scientific and Technological Innovation Project of China Academy of Chinese Medical Sciences [CI2021B003], and the Fundamental Research Funds for the Central Public Welfare Research Institutes of China [Z0735].

## Author contributions

Conceptualization: P.W., Q.W., D.W., and W.Y., Methodology: K.C., Z.F., R.S., Q.Y., and F.W., Formal analysis: K.C., Z.F., and X.S., Investigation: K.C. R.S., Q.Y., Y.T., and Y.X., Resources: P.W. and D.W., Data curation: F.W., X.S., Writing original draft: K.C. and Z.F., Writing review & editing: P.W., Q.W. and W.Y., Visualization: Y.T. and Y.X., Supervision: P.W., Q.W., D.W., and W.Y., Project administration: D.W., Funding acquisition: P.W. and W.Y.

## Competing interests

Qing Wei, Weifeng Yang and Kaiyu Cui are joint inventors of a patent filed with the China National Intellectual Property Administration (No.202511090940.9), covering the application of OxyR regulon as antibacterial targets.
