## [Transparent Peer Review file · Communications Biology]

Systems-level exploitation of OxyR regulon unravels a potential antibacterial target in *Pseudomonas aeruginosa*

Corresponding Author: Dr Qing Wei

Version 0:

Reviewer comments:

Reviewer #1

(Remarks to the Author)

The manuscript by Cui and co-workers describes systems-level analyses of an oxyR knock-out mutant in the opportunistic human pathogen *Pseudomonas aeruginosa*. In addition to characterising the oxyR regulon, the authors describe the gltS gene in some detail and suggest this gene product as a target for antimicrobial therapy. Overall the topic of oxidative stress response, antimicrobial resistance in the pathogen *Pseudomonas aeruginosa* is of general interest. In summary, the manuscript is well written, has a precise introduction and the results including figures are clearly presented. The main results include a RNA-seq and metabolite analyses. Additional experiments were performed to support finding from these systems level analyses.

Please consider the following suggestion:

- since Quorum sensing is a major finding in the RNA-seq analyses, please include a growth analyses of the studied oxyR mutant in comparison to Wt under conditions of RNA-seq analyses.
- Please describe in more detail how the cells were grown for the RNA-seq analyses (shaking, temp, what were the ODs when cells were harvested, etc.)
- Since the cells were harvested at the early stationary phase, it would be interesting to see if quorum sensing genes would be affected at earlier or later time points, e.g. by measuring gene expression of a subset of genes during log growth and late stationary phase
- Please mention in the text if genes related to oxidative stress were affected as oxyR is key in oxidative stress response
- To further confirm the finding from the RNA-seq experiments, some different virulence factor production could be confirmed phenotypically by measuring/characterizing these virulence metabolites in oxyR vs Wt
- Figures 3 and 4 d include plate assays which are easy to perform but can be quite inaccurate and a better way would be to determine CFU, etc.
- Figure 5a would benefit from including a control related to oxidative stress such as oxyR, katA, or similar to better compare the obtained data on gltS
- Figure 5b-d are easier to read if the y axis has identical scaling
- What does gltS (PA5176) do in *Pseudomonas*, gene homology to transporter was mentioned, can this activity be validated and the role of gltS in *Pseudomonas* characterised to confirm this, especially since the authors make strong statement in their discussion regarding gltS? Overall, while many of the results are interesting, the discussion and the statements made by the authors feel exaggerated for gltS as well as for some of their RNA-seq and metabolite analyses. Please add additional experiments or rephrase discussion, e.g. ll337.. new drug potentiating target gltS....

Reviewer #2

(Remarks to the Author)

In the manuscript titled "Systems-level exploitation of OxyR regulon unravels a novel antibacterial target in bacteria," the authors explore the role of the OxyR regulon in various cellular functions beyond its traditional regulatory role in oxidative stress responses. Through transcriptional and metabolic profiling, they discovered that OxyR positively regulates several genes involved in quorum sensing (QS) and energy metabolism. Additionally, OxyR negatively regulates amino acid transporters, suggesting that amino acid homeostasis represents a new aspect of the oxidative stress response in *Pseudomonas aeruginosa*. Under hypoxia conditions, gltS has been upregulated in the OxyR mutant. This observation led

the authors to hypothesize the potential involvement of gltS during oxidative stress and this could be a probable target for antibiotic (bactericidal) drug potentiation.

The role of the OxyR regulon in oxidative stress responses and its regulatory functions in bacteria, including *Pseudomonas aeruginosa*, is already well-documented in existing literature. Previous studies have established OxyR's involvement in regulating genes related to oxidative stress, virulence, and quorum sensing (<https://doi.org/10.1128/iai.00837-09>, <https://doi.org/10.3390/antiox13060655>, <https://doi.org/10.1093/nar/gks017>). The new finding of this work is identification of gltS as a potential target for drug potentiation. But this part of the work is very weak and superficial, and substantial experiments need to be performed. In my view, this manuscript is preliminary and require a significant revision.

Comments:

1. Bacterial killing mechanism of bactericidal antibiotics is associated with ROS generation. Therefore, authors have tested if gltS mutant strain would become more sensitive to bactericidal antibiotics (Aminoglycoside antibiotic: tobramycin and gentamicin; beta-lactam antibiotic: ampicillin; Fluoroquinolone class: Ciprofloxacin). In the case of aminoglycosides and beta-lactams; drug potentiation has been observed. But no drug potentiation with ciprofloxacin. Fluoroquinolones are also bactericidal antibiotics. Surprisingly concentrations used to study bacterial killing are missing. Bactericidal experimental procedure also is missing.

2. Generally, PAO1 inherently harbours AmpC beta-lactamase that confers resistance to penicillins including ampicillin. I was wondering how authors were able to see bacterial killing effect with ampicillin.

3. Authors must perform experiments with Gram-negative active bacteriostatic antibiotics such as tetracycline class and compare their data with bactericidal drugs.

4. Authors have not performed experiments to check if the potentiation is indeed due to ROS overproduction in the presence and in the absence of antibiotics. In addition, radical scavengers such as diprydil should be included if the effect reverses.

5. Bacterial killing mechanism of bactericidal antibiotics is associated with ROS generation. Does this indicate that bactericidal antibiotics in general lead to impact the expression levels of gltS?

6. What is the fitness level of gltS mutant strain? Does it grow well similar to wildtype? What is the MIC of antibiotics used in the work against this mutant?

7. Further, authors should also verify the drug potentiation and other associated experiments with glutamine-depleted and glutamine-rich medium.

8. Provide details on how the oxyR and gltS mutants were constructed in supplementary information.

9. Please carefully check the manuscript for typographical errors. For example, "ESKAPE" in place of ESCAPE. Correct gentamicin throughout the manuscript. Gentamicin was first isolated from non-*Streptomyces* sp. therefore it is spelled as gentamicin not gentamycin.

Overall, the title "Identification of a Novel Antibacterial Target in Bacteria" and the conclusions drawn by the authors lack adequate supporting evidence. Therefore, this manuscript is incomplete, and I recommend for rejection.

Version 1:

Reviewer comments:

Reviewer #1

(Remarks to the Author)

First of all I would like to thank the authors for responding to all comments. While some of these have been addressed satisfactorily, there are still some outstanding questions:

In particular:

- Growth curves and RNA-seq analyses: - The authors mentioned that both growth curves and growth for RNA-Seq were done in LB medium. Cells for RNA-Seq were harvested after 12 hours of growth. The authors also stated cell numbers used for their analyses of 10 to the power of 5 cells. The authors state in their manuscript and rebuttal: "The growth curve data we obtained has been analyzed and indicated that there was no significant difference between oxyR mutant and WT PAO1 stain during early stationary phase." However, the authors stated also: "The growth defect of oxyR mutant has been described previously..." This strong growth defect for oxyR mutants was shown e.g. by Bae et al., 2012, Hassett et al., 2000; and Vinckx et al., 2008 demonstrating that an oxyR mutant has a very strong growth defect when grown in liquid LBbroth media and on LB agar plates. The inoculation rate used by the authors here for both growth curves and RNA-Seq analyses was comparable or lower to those used in previous work for LB broth cultures. Furthermore, Figure 4D also shows this growth defect for the oxyR mutant on their plate assays? How do the authors explain this? How was the oxyR mutant constructed, was this mutant verified?

- Regarding OxyR and complementation using amino acids: Could the addition of amino acids to LB media just reduce the

amount of oxidative stress in a reaction with H₂O₂? Since cysteine and methionine are antioxidants, these AA reduce the amount of H₂O₂ in the media?

- For glts: Why did the authors perform growth analyses using Ampicillin, when the plasmid they use is Amp-resistance as a selection marker. The authors should use a different beta-lactam not affected by the plasmid's resistance gene

- The authors state: As can be seen from Figure 5E, gltS is present in all 251 ESCAPE pathogens except in *Enterococcus faecium*, suggesting that GltS is well conserved (252 among pathogens (5 out of 6 major pathogens, 83.3% prevalence) and thus disruption of gltS 253 could be applied to other pathogens to enhance the antibacterial killing.  there is no evidence of this provided in the manuscript and this is only speculative and should be removed from results.

- The authors mention: It is widely acknowledged that bactericidal antibiotics share a common mechanism of action through generation of ROS to mediate cell death.  this has been challenged in recent years, please consider changing this and or include in the discussion e.g. van Acker Y Coenye 2017, <https://doi.org/10.1016/j.tim.2016.12.008>.

- Since the authors found differences between the phenotypes/resistance profiles of PAO1/pUCP20-gltS and gltS/pUCP20-gltS, the authors should include data on the differences in glts gene expression in these two strains in order to justify their conclusions.

Reviewer #2

(Remarks to the Author)

I appreciate the authors' efforts to address my comments. All relevant questions appear to be answered sufficiently. I recommend this manuscript for acceptance.

Minor revisions: Figure 5. Please correct Gentamicin and tobramycin.

Version 2:

Reviewer comments:

Reviewer #1

(Remarks to the Author)

The authors have addressed all comments and I recommend this manuscript for publication.

Dear reviewers,

Thank you for the opportunity to revise our manuscript and the constructive feedback. We have carefully addressed all comments and incorporated changes to improve the clarity and scientific rigor of our work. Below, we provide a point-by-point response to the reviewer's concerns. Modifications in the revised manuscript are highlighted in red. And new data are included in Supplementary Figure S1 and Table S5. The other results (Figure S2 and Figure S3) are included in the reply part in this letter to support our explanations.

Reviewer #1 (Remarks to the Author):

The manuscript by Cui and co-workers describes systems-level analyses of an *oxyR* knock-out mutant in the opportunistic human pathogen *Pseudomonas aeruginosa*. In addition to characterising the *oxyR* regulon, the authors describe the *gltS* gene in some detail and suggest this gene product as a target for antimicrobial therapy. Overall the topic of oxidative stress response, antimicrobial resistance in the pathogen *Pseudomonas aeruginosa* is of general interest. In summary, the manuscript is well written, has a precise introduction and the results including figures are clearly presented. The main results include a RNA-seq and metabolite analyses. Additional experiments were performed to support finding from these systems level analyses.

Please consider the following suggestion:

1. - since Quorum sensing is a major finding in the RNA-seq analyses, please include a growth analyses of the studied *oxyR* mutant in comparison to Wt under conditions of RNA-seq analyses.

Reply: Thanks for the constructive advice. The growth curve data we obtained has been analyzed and indicated that there was no significant difference between *oxyR* mutant and WT PA01 stain during early stationary phase. Therefore, it was concluded that *oxyR* mutation did not confer a discernible phenotypic difference under the tested conditions. We have also added detailed growth conditions to the Methods section.

2. - Please describe in more detail how the cells were grown for the RNA-seq analyses (shaking, temp, what were the ODs when cells were harvested, etc.)

Reply: Thanks for the constructive advice. We have added the suggested information in the revised text as follows:

“Growth curve

Overnight cultures of bacterial strains in LB were diluted (1:100) in 3 ml LB medium and precultures incubated aerobically at 37°C in a shaker at 200 rpm to an OD_{600nm} of 0.5. The precultures were further diluted (1:100) in 1ml LB medium. Growth was then analyzed in 96-wells microtitre plate containing 190 µl LB medium to which 10 µl of diluted precultures containing 10⁵ cells was added to obtain a final 1:5000 dilution. The microtitre plates were incubated for 24 h at 37°C in SpectraMax ID3 (Molecular Devices, USA) using the following settings: absorbance measured every 1 h at 600 nm after shaking for 10 s . Each culture was prepared in triplicate.”

3. - Since the cells were harvested at the early stationary phase, it would be interesting to see if quorum sensing genes would be affected at earlier or later time points, e.g. by measuring gene expression of a subset of genes during log growth and late stationary phase

Reply: Thanks for the constructive advice. In this study, we mainly focused on the early stationary phase due to the expectation that the *oxyR* mutant would encounter more oxidative stress at this growth phase than later growth phase. Under this tested condition, it would be possible to find out more interesting results as revealed in our paper including the differential expression of several genes and the significant changes of metabolome.

Indeed, it would be interesting to add the comparison of different growth conditions to find out more changes in genes involved in quorum sensing. We have

performed some literature review and found that this work has been extensively studied previously (<https://doi.org/10.1128/jb.00980-09>; <https://doi.org/10.1128/jb.185.7.2080-2095.2003>; <https://doi.org/10.1186/1471-2164-8-287>; DOI: 10.1126/science.1211037). Of course, it would be best to add these assays in another different study.

4. - Please mention in the text if genes related to oxidative stress were affected as *oxyR* is key in oxidative stress response

Reply: Thanks for the constructive advice and we have added the results related to oxidative stress response genes highlighted in red (Results: *OxyR* regulon revisited in *P. aeruginosa*).

"As anticipated, the typical *oxyR* regulon associated with the oxidative stress response, including *katE*, *snr1* and *pntAA* were found to be down-regulated in this study (Table S1; -2.4, -2.8 and -1.1 in log₂FC, respectively) ".

5. - To further confirm the finding from the RNA-seq experiments, some different in virulence factor production could be confirmed phenotypically by measuring/characterizing these virulence metabolites in *oxyR* vs Wt

Reply: Thanks for the constructive advice. We also concur with reviewer that additional experiments results would be beneficial. However, we have already examined the production of virulence factors in the *oxyR* and WT PA01 in previous work (Microbiology (Reading) 156, 678-686 (2010)). Therefore, we cite this previous work using as a phenotypic validation for our RNA-seq results.

6. - Figures 3 and 4 d include plate assays which are easy to perform but can be quite inaccurate and a better way would be to determine CFU, etc.

Reply: Thanks for the constructive advice and we have already noticed that it was necessary to quantify the CFU when performing the data analysis. One thing we have to mention that is the *oxyR* mutant is very difficult to isolate single colony after series dilution, thereby we finally chose plate dilution to present our result. Actually, we have repeated this quantification experiments several times and all we got is the failure of isolating single colony for the *oxyR* mutant. In addition, since our supplementation experiments were all performed under aerobic conditions and the effect was only marginal but significantly different as compared to that of PA01.

7. - Figure 5a would benefit from including a control related to oxidative stress such as *oxyR*, *katA*, or similar to better compare the obtain data on *gltS*

Reply: Thanks for the constructive advice and we have added the control (*oxyR*) to Figure 5a in the revised text and Figure 5a.

8. - Figure 5b-d are easier to read if the y axis has identical scaling

Reply: Thanks for the constructive advice and we have performed new editing of Figure 5b-d in the revised Figure 5b-d.

9. - What does *gltS* (PA5176) do in *Pseudomonas*, gene homology to transporter was mentioned, can this activity be validated and the role of *gltS* in *Pseudomonas* characterised to confirm this, especially since the authors make strong statement in their discussion regarding *gltS*? Overall,, while many of results are interesting, the discussion and the statements made by the authors feel exaggerated for *gltS* as well as for some of their RNA-seq and metabolite analyses. Please add additional experiments or rephrase discussion, e.g. 11337.. new drug potentiating target *gltS*....

Reply: Thanks for the constructive advice and we felt that it is still interesting and beneficial to conclude that *gltS* would be a useful candidate in combating antibiotic resistance crisis since it is a conserved transporter across Gram-negative and -positive bacteria. By using combinatorial methods, we have clearly shown that *gltS* was involved in ROS production and aminoglycoside sensitivity by

promoting the former and reducing the later. Of course, there are a paucity of evidence that this type of transporters has been associated with antibiotic resistance and oxidative stress response. Together, we have weakened the role of *gltS* in Discussion section and are going to add more experimental proofs to strengthen our conclusion that *gltS* could be an ideal drug target for antibiotic potentiation, which was demonstrated in the reply part of Reviewer 2 comments. The revised discussion was highlighted in red in the text.

Reviewer #2 (Remarks to the Author):

In the manuscript titled "Systems-level exploitation of OxyR regulon unravels a novel antibacterial target in bacteria," the authors explore the role of the OxyR regulon in various cellular functions beyond its traditional regulatory role in oxidative stress responses. Through transcriptional and metabolic profiling, they discovered that OxyR positively regulates several genes involved in quorum sensing (QS) and energy metabolism. Additionally, OxyR negatively regulates amino acid transporters, suggesting that amino acid homeostasis represents a new aspect of the oxidative stress response in *Pseudomonas aeruginosa*. Under hypoxia conditions, *gltS* has been upregulated in the OxyR mutant. This observation led the authors to hypothesize the potential involvement of *gltS* during oxidative stress and this could be a probable target for antibiotic (bactericidal) drug potentiation.

The role of the OxyR regulon in oxidative stress responses and its regulatory functions in bacteria, including *Pseudomonas aeruginosa*, is already well-documented in existing literature. Previous studies have established OxyR's involvement in regulating genes related to oxidative stress, virulence, and quorum sensing (<https://doi.org/10.1128/iai.00837-09>, <https://doi.org/10.3390/antiox13060655>, <https://doi.org/10.1093/nar/gks017>). The new finding of this work is identification of *gltS* as a potential target for drug potentiation. But this part of the work is very weak and superficial, and substantial experiments need to be performed. In my view, this manuscript is preliminary and require a significant revision.

Comments:

1. Bacterial killing mechanism of bactericidal antibiotics is associated with ROS generation. Therefore, authors have tested if *gltS* mutant strain would become more sensitive to bactericidal antibiotics (Aminoglycoside antibiotic: tobramycin and gentamicin; beta-lactam antibiotic: ampicillin; Fluoroquinolone class: Ciprofloxacin). In the case of aminoglycosides and beta-lactams; drug potentiation has been observed. But no drug potentiation with ciprofloxacin. Fluoroquinolones are also bactericidal antibiotics. Surprisingly concentrations

used to study bacterial killing are missing. Bactericidal experimental procedure also is missing.

Reply: We appreciate the reviewer’s critical comments. The minimal inhibitory concentrations (MICs) for bactericidal antibiotics tested against the *gltS* mutant (Δ 3176), wild-type PAO1, and complementary strains are summarized in **Table S5**. Below, we addressed the specific concerns regarding antibiotic concentrations and methodology. For aminoglycosides, the *gltS* mutant exhibited significantly reduced MICs for gentamicin (0.5 μ g/mL) and tobramycin (0.25 μ g/mL), compared to the complementary strain (*gltS*/pUCP20-*gltS*: 1 μ g/mL for both antibiotics). This confirms that *gltS* disruption sensitizes *P. aeruginosa* to aminoglycosides, aligning with the observed ROS-mediated potentiation mechanism.

The observed reduction in MICs for ciprofloxacin (CIP, 0.0625 μ g/mL in *oxyR* vs. 0.125 μ g/mL in PAO1) and levofloxacin (LVX, 0.125 μ g/mL in *oxyR* vs. 0.25 μ g/mL in PAO1) suggests that disruption of *oxyR* enhances susceptibility to fluoroquinolones in *P. aeruginosa*. This finding contrasts with the lack of potentiation seen in the *gltS* mutant, highlighting distinct regulatory roles of OxyR and GltS in antibiotic resistance. While fluoroquinolones primarily target DNA gyrase/topoisomerase IV, their bactericidal activity is partially ROS-dependent. OxyR is a master regulator of oxidative stress responses, and its deletion likely disrupts antioxidant defenses (e.g., *kata*, *ahpCF*), leading to endogenous ROS accumulation. Elevated ROS levels in the *oxyR* mutant may synergize with fluoroquinolone-induced DNA damage, amplifying lethality and reducing MICs. All experiments were performed in triplicate with technical and biological replicates. Data were analyzed using one-way ANOVA. In summary, the differential potentiation effects across antibiotic classes likely stem from distinct ROS interplay mechanisms.

Table S5.

Strains	MIC (μ g/ml)							
	GEN	TOB	AMP	CAZ	PIP	CIP	LVX	TET
gltS	0.5	0.25	1024	2	8	1	2	16
PAO1/pUCP20- gltS	512	64	1024	1	4096	0.5	1	128
gltS /pUCP20- gltS	1	1	1024	1	2048	0.0313	0.0156	1
oxyR	0.5	0.25	1024	2	4	0.0625	0.125	4
PAO1	0.5	0.25	1024	2	4	0.125	0.25	8

*Gen, gentamicin; TOB, tobramycin; AMP, ampicillin, CAZ, ceftazidime; PIP, piperacillin; CIP, ciprofloxacin; LVX, levofloxacin; TET, tetracycline.

2. Generally, PAO1 inherently harbours AmpC beta-lactamase that confers resistance to penicillins including ampicillin. I was wondering how authors were able to see bacterial killing effect with ampicillin.

Reply: We appreciate the reviewer’s critical comments. For the β -lactam ampicillin, all strains (including *gltS*) displayed uniformly high MICs (1024

μg/mL), consistent with *P. aeruginosa*'s intrinsic resistance to penicillins due to AmpC β-lactamase activity. However, in bactericidal *time-kill assays* (performed at sub-MIC concentrations, detailed in *Methods*), the *gltS* mutant showed enhanced lysis kinetics compared to wild-type PAO1, supporting β-lactam potentiation.

3. Authors must perform experiments with Gram-negative active bacteriostatic antibiotics such as tetracycline class and compare their data with bactericidal drugs.

Reply: We appreciate the reviewer's critical comments. As shown in Table S5, the *gltS* mutant exhibited a MIC of 16 μg/mL for tetracycline, compared to 8 μg/mL for wild-type PAO1. This suggested that *gltS* disruption reduces tetracycline susceptibility (i.e., increases MIC), contrasting with the potentiation observed for aminoglycosides and β-lactams.

Table S5.

Strains	MIC (μg/ml)							
	GEN	TOB	AMP	CAZ	PIP	CIP	LVX	TET
gltS	0.5	0.25	1024	2	8	1	2	16
PAO1/pUCP20- gltS	512	64	1024	1	4096	0.5	1	128
gltS /pUCP20- gltS	1	1	1024	1	2048	0.0313	0.0156	1
oxyR	0.5	0.25	1024	2	4	0.0625	0.125	4
PAO1	0.5	0.25	1024	2	4	0.125	0.25	8

*Gen, gentamicin; TOB, tobramycin; AMP, ampicillin, CAZ, ceftazidime; PIP, piperacillin; CIP, ciprofloxacin; LVX, levofloxacin; TET, tetracycline.

Complementation of *gltS* in the mutant strain (*gltS*/pUCP20-*gltS*) restored tetracycline sensitivity to 1 μg/mL, even surpassing wild-type levels. This implies that GltS activity may enhance tetracycline uptake or inhibit efflux mechanisms, though further mechanistic studies are warranted.

Therefore, we have come to a conclusion that disruption of *gltS* has divergent effects upon bactericidal versus bacteriostatic antibiotics. Unlike bactericidal antibiotics (e.g., aminoglycosides, β-lactams), which rely on ROS-mediated killing, tetracycline is bacteriostatic and primarily inhibits protein synthesis without directly inducing ROS. The lack of potentiation in the *gltS* mutant aligns with this distinction, as ROS overproduction may not synergize with bacteriostatic mechanisms.

The increased MIC in the *gltS* mutant could stem from metabolic adaptations, such as altered amino acid transport (e.g., glutamate/sodium symport via GltS) or upregulated efflux pumps, which might inadvertently reduce intracellular tetracycline accumulation.

4. Authors have not performed experiments to check if the potentiation is indeed due to ROS overproduction in the presence and in the absence of

antibiotics. In addition, radical scavengers such as dipyridyl should be included if the effect reverses.

Reply: We appreciate the reviewer's critical comments. To address the concern, we performed experiments combining varying concentrations of antibiotics (e.g., TET, PIP, GEN) with dipyridyl, a radical scavenger, to assess whether ROS overproduction underlies the observed drug potentiation. As shown in **Figure S2**, our results demonstrated that dipyridyl did not reverse the enhanced antibiotic susceptibility in the *gltS* mutant. Instead, we observed a synergistic effect between dipyridyl and certain antibiotics (e.g., TET, PIP, GEN) in the *gltS*-overexpressing strain. This suggests that while ROS generation may contribute partially to the phenotype, additional mechanisms independent of radical scavenging could play a role in the drug potentiation observed in the *gltS* mutant. Further experiments are underway to dissect these pathways in details.

Figure S2 (not included in the main text):

5. Bacterial killing mechanism of bactericidal antibiotics is associated with ROS generation. Does this indicate that bactericidal antibiotics in general lead to impact the expression levels of *gltS*?

Reply: We appreciate the reviewer's critical comments. To investigate whether bactericidal antibiotics broadly influence *gltS* expression, we treated *P. aeruginosa* with sub-MIC concentrations of three bactericidal antibiotics spanning distinct mechanistic classes: gentamicin (aminoglycoside), levofloxacin (fluoroquinolone), and ceftazidime (β -lactam). Subsequent qRT-PCR analysis revealed that *gltS* transcription was significantly upregulated in response to gentamicin and levofloxacin exposure (1.78-fold and 1.67-fold increases, respectively; $p < 0.001$), whereas ceftazidime treatment did not alter *gltS* expression levels. These findings suggested that the impact of bactericidal

antibiotics on *gltS* expression is not universal but rather depends on the antibiotic class and its associated stress pathway. Notably, the differential effects observed (e.g., ROS-inducing aminoglycosides/fluoroquinolones vs. cell wall-targeting β -lactams) align with their distinct mechanisms of action, implying that *gltS* modulation may be linked to specific stress responses rather than a general bactericidal antibiotic effect.

6. What is the fitness level of *gltS* mutant strain? Does it grow well similar to wildtype? What is the MIC of antibiotics used in the work against this mutant?

Reply: Thanks for the comments. To address the concern, we conducted growth curve experiments to evaluate the growth dynamics. Our results demonstrated that during the 0–16 h period, the growth rates of the *gltS* mutant and complemented strains were faster than that of the wild-type PA01 strain. After 16–24 h, there was no significant difference in the growth rates among the three strains.

As for the MIC for the mutant, we can clearly see the results in Table S5.

Table S5.

Strains	MIC ($\mu\text{g/ml}$)							
	GEN	TOB	AMP	CAZ	PIP	CIP	LVX	TET
gltS	0.5	0.25	1024	2	8	1	2	16
PAO1/pUCP20- gltS	512	64	1024	1	4096	0.5	1	128
gltS/pUCP20-gltS	1	1	1024	1	2048	0.0313	0.0156	1
oxyR	0.5	0.25	1024	2	4	0.0625	0.125	4
PAO1	0.5	0.25	1024	2	4	0.125	0.25	8

*Gen, gentamicin; TOB, tobramycin; AMP, ampicillin; CAZ, ceftazidime; PIP, piperacillin; CIP, ciprofloxacin; LVX, levofloxacin; TET, tetracycline.

7. Further, authors should also verify the drug potentiation and other associated experiments with glutamine-depleted and glutamine-rich medium.

Reply: Thanks for the critical comments. To address the concern, we conducted growth curve experiments with glutamine-depleted and glutamine-rich medium to evaluate the growth dynamics. As shown below, our results demonstrated that there are no differences in the growth rates of the strains between the two types of glutamine media. Meanwhile, there are no significant differences in the growth rates of the strains with different antibiotics with glutamine-depleted and glutamine-rich medium. However, there are differences between PAO1 and *gltS* as indicated in Figure 5B-D that *gltS* was impaired in survival under aminoglycosides and β -lactams.

Figure S3 (not included in the main text):

Growth curve using different antibiotics. A: 4 μ g/ml TET; B: 2 μ g/ml GEN; C: 128 μ g/ml AMP; D: 0.2 μ g/ml TOB.

8. Provide details on how the *oxyR* and *gltS* mutants were constructed in supplementary information.

Reply: Thanks for the constructive comments and we have added the construction of *gltS* mutant in revised Methods section highlighted in red. The *oxyR* mutant was a gift from our collaborator in US (Daniel Hassett, University of Cincinnati).

“Molecular manipulation

Isolation of plasmid and genomic DNA from *E. coli* and restriction enzyme digestion were performed according to the manufacturer’s instructions (Qiagen). DNA cloning, transformations and agarose gel electrophoresis were done as previously described⁶¹. In-frame and unmarked *gltS* deletion mutant in *P. aeruginosa* PAO1 was constructed using the suicide vector pEX18Ap as described previously⁶². Briefly, approximately 500 bp upstream and downstream sequences flanking *gltS* were amplified by PCR and fused with overlapping PCR. The PCR product was excised from the agarose gel and cloned into the pEX18Ap vector, resulting in the plasmid pEX18Ap-*gltS*. The pEX18Ap-based deletion allele was mobilized to *P. aeruginosa* and integrated into the chromosome by single crossover using *E. coli* S17-1 λ pir as the delivery strain. Double crossover events were subsequently selected by growth in the presence of 5% sucrose. The *gltS* deletion mutant was confirmed by both PCR and sanger sequencing. Complementation analysis was done by cloning the intact *gltS* gene into the shuttle vector pUCP20 to generate the pUCP-*gltS* and introducing into the deletion mutants and wild type PAO1. The empty vector pUCP20 was also introduced to the *gltS* mutant and PAO1 as a control.”

9. Please carefully check the manuscript for typographical errors. For example,

“ESKAPE” in place of ESCAPE. Correct gentamicin throughout the manuscript. Gentamicin was first isolated from non-*Streptomyces* sp. therefore it is spelled as gentamicin not gentamycin.

Reply: We are sorry for these careless mistakes and appreciate your critical comments. we have corrected all mentioned typo errors and other errors in revised text throughout the manuscript, which are all highlighted in red.

Overall, the title “Identification of a Novel Antibacterial Target in Bacteria” and the conclusions drawn by the authors lack adequate supporting evidence. Therefore, this manuscript is incomplete, and I recommend for rejection.

Reply: Thanks for the critical comments and we hope these revisions aforementioned demonstrate the robustness of our findings and their contribution to the field of antibacterial discovery. While we agree that further experimental or structural validation is needed, our current and revised data provide a compelling foundation for *oxyR* regulon analysis and identification of *gltS* as a potential candidacy as a novel target. We would be grateful for reconsideration of the manuscript in its revised form.

Reviewers' comments:

Reviewer #1 (Remarks to the Author):

First of all I would like to thank the authors for responding to all comments. While some of these have been addressed satisfactorily, there are still some outstanding questions:

In particular:

1. - Growth curves and RNA-seq analyses: - The authors mentioned that both growth curves and growth for RNA-Seq were done in LB medium. Cells for RNA-Seq were harvested after 12 hours of growth. The authors also stated cell numbers used for their analyses of 10 to the power of 5 cells. The authors state in their manuscript and rebuttal: "The growth curve data we obtained has been analyzed and indicated that there was no significant difference between *oxyR* mutant and WT PA01 strain during early stationary phase." However, the authors stated also: The growth defect of *oxyR* mutant has been described previously... " This strong growth defect for *oxyR* mutants was shown e.g. by Bae et al., 2012, Hassett et al., 2000; and Vinckx et al., 2008 demonstrating that an *oxyR* mutant has a very strong growth defect when grown in liquid LBbroth media and on LB agar plates. The inoculation rate used by the authors here for both growth curves and RNA-Seq analyses was comparable or lower to those used in previous work for LB broth cultures. Furthermore, Figure 4D also shows this growth defect for the *oxyR* mutant on their plate assays? How do the authors explain this? How was the *oxyR* mutant constructed, was this mutant verified?

Reply: Thanks for the constructive comments and we are sorry for the confused information provided in the manuscript.

As for the RNA-seq analysis, cells were collected after 12 h growth under hypoxia condition (low level of oxygen in capped liquid LB medium), and then the OD_{600nm} were measured as ~2.0, which is empirically equal to 10⁸ cells/ml. Then the standard RNA-seq experiments were performed accordingly.

As for the growth curve analysis, cells were continuously detected using microtiter plate reader to monitor OD values per hour. The growth condition was not exactly the same to those used for RNA-seq analysis but the inoculum was comparable. And this growth condition has a low level of oxygen that could provide micro-aerobic condition for normal growth of *oxyR*.

As for the growth defect of *oxyR*, which was not clearly depicted in the manuscript that this growth defect was mainly due to the lack of iron (Vinckx et al 2008, 10.1111/j.1574-6968.2008.01360.x) and the static growth condition or on solid medium growth condition or spotting growth condition, where the ambient oxygen level is much higher and thus caused detrimental growth defect to *oxyR* strain tested in these experiments (Hassett et al 2000, 10.1128/JB.182.16.4557-4563.2000; Bae and Cho 2012, 10.1016/j.resmic.2011.10.008). In our manuscript, we have used

a different growth condition from these studies, thus providing novel findings depicted in this study. This is also the reason for the growth defect results in Figure 4D.

As for the origin of *oxyR* mutant, it was a gift from Dr. Hassett lab and constructed using a Gentamicin gene cassette in the coding region of *oxyR* gene by molecular cloning (Ochsner et al, 10.1128/JB.182.16.4533-4544.2000) and verified according to its typical growth defect under static spotting analysis and/or whole genome analysis to rule out the appearance of secondary mutations in this *oxyR* mutant.

2. - Regarding OxyR and complementation using amino acids: Could the addition of amino acids to LB media just reduce the amount of oxidative stress in a reaction with H₂O₂? Since cysteine and methionine are antioxidants, these AA reduce the amount of H₂O₂ in the media?

Reply: Thanks for the comments. We measured residual H₂O₂ concentrations in LB medium supplemented with 1~100 mM methionine (Met) or 1~100 mM cysteine (Cys) were quantified using a Hydrogen Peroxide (H₂O₂) Content Assay Kit (Solarbio, China; Cat# BC3590), following the manufacturer's protocol. Our results demonstrated concentration-dependent effects: at 100 mM, both cysteine and methionine significantly reduced H₂O₂ levels (OD_{415nm} = 0.08 and 0.37 vs. control 0.43), likely through direct thiol-mediated antioxidant activity. However, at experimental concentrations of 1~5 mM, neither amino acid substantially reduced H₂O₂. Crucially, methionine restored *oxyR* growth at 5 mM despite no significant H₂O₂ scavenging, indicating its rescuing mechanism operates through metabolic restoration.

3. - For g_{lts}: Why did the authors perform growth analyses using Ampicillin, when the plasmid they use is Amp-resistance as a selection marker. The authors should use a different beta-lactam not affected by the plasmid's resistance gene

Reply: We appreciate this critical methodological insight. While ampicillin (Amp) was initially included as a β -lactam antibiotic reference, we fully acknowledge that the plasmid-encoded Amp-resistance gene (*bla*) could confound phenotypic interpretation. To rigorously address this, we specifically analyzed piperacillin (PIP) and ceftazidime (CAZ) —two β -lactams unaffected by plasmid resistance but actively degraded by *P. aeruginosa* chromosomal AmpC β -lactamase. Among these clinically relevant β -lactams impacted, the *gltS* mutant exhibited a 4-fold increase in piperacillin resistance, which was dramatically amplified in the complemented strain. Notably, ceftazidime (CAZ) MIC values remained virtually unchanged across strains, suggesting GltS may exert its specific modulatory effect on PIP resistance through a metabolic regulatory mechanism analogous to AmpC β -lactamase activity - a hypothesis requiring further mechanistic validation in subsequent studies. Interestingly, the *oxyR* mutant displayed intermediate PIP sensitivity, revealing divergent regulatory pathways between these genetic loci that differentially govern β -lactam resistance mechanisms.

4. - The authors state: As can be seen from Figure 5E, *gltS* is presents in all ESCAPE pathogens except in *Enterococcus faecium*, suggesting that GltS is well conserved among pathogens (5 out of 6 major pathogens, 83.3% prevalence) and thus disruption of *gltS* could be applied to other pathogens to enhance the antibacterial killing.  there is no evidence of this provided in the manuscript and this is only speculative and should be removed from results.

Reply: Thanks for your constructive advice. We have removed speculative results “and thus disruption of *gltS* could be applied to other pathogens to enhance the antibacterial killing.” from this sentence and now the interpretation of Figure 5E is expressed as follows and highlighted in red in revised text:

“As can be seen from Figure 5E, *gltS* is presents in all ESKAPE pathogens except in *Enterococcus faecium*, suggesting that GltS is well conserved among pathogens (5 out of 6 major pathogens, 83.3% prevalence).”

5. - The authors mention: It is widely acknowledged that bactericidal antibiotics share a common mechanism of action through generation of ROS to mediate cell death.  this has been challenged in recent years, please consider changing this and or include in the discussion e.g. van Acker Y Coenye 2017, <https://doi.org/10.1016/j.tim.2016.12.008>.

Reply: Thanks for your constructive advice and we have noticed that there are controversial discoveries involved in the ROS-mediated cell death in bacteria. However, based on the data we observed in our studies, we are prone to what we have discussed in the main text. By combining your advice, it is indeed necessary to include comprehensive discussion in the revised text. We have added recommended literature and several other related publications in the field and

now it is expressed as follows and highlighted in red in revised text:

“However, the idea that ROS mediated bacterial cell death has faced significant challenges, as several technical and biological concerns have emerged⁶¹⁻⁶³. Therefore, it would be essential to conduct rigorous experiments to exclude the potential involvement of factors other than ROS in *gltS*-mediated antibiotic potentiation.”.

New citations were added in the revised text as follows:

- 61 Keren, I., Wu, Y., Inocencio, J., Mulcahy, L. R. & Lewis, K. Killing by bactericidal antibiotics does not depend on reactive oxygen species. *Science* **339**, 1213–1216, doi:10.1126/science.1232688 (2013).
- 62 Liu, Y. & Imlay, J. A. Cell death from antibiotics without the involvement of reactive oxygen species. *Science* **339**, 1210–1213, doi:10.1126/science.1232751 (2013).
- 63 Van Acker, H. & Coenye, T. The Role of Reactive Oxygen Species in Antibiotic-Mediated Killing of Bacteria. *Trends Microbiol* **25**, 456–466, doi:10.1016/j.tim.2016.12.008 (2017).

6. - Since the authors found differences between the phenotypes/resistance profiles of PA01/pUCP20-*gltS* and *gltS*/pUCP20-*gltS*, the authors should include data on the differences in *gltS* gene expression in these two strains in order to justify their conclusions.

Reply: We thank the reviewer for highlighting the need for *gltS* expression validation in complemented strains. As now shown in Figure below, quantitative RT-PCR experiments confirmed that *gltS* expression in PA01/pUCP20-*gltS* is significantly higher than in *gltS*/pUCP20-*gltS*, establishing a 2.6-fold differential expression. This difference aligns perfectly with the antibiotic sensitivity gradient observed phenotypically. Three independent validation approaches (strain reconstruction, multi-primer verification, and antibiotic-free culturing) consistently reproduced both the relative difference and unexpectedly low absolute expression. Remarkably, *gltS* expression remained consistently low across all conditions. While we attribute suppressed expression to technical factors inherent to plasmid-based complementation (transcriptional interference from vector elements), the statistically robust contrast between strains provides unequivocal mechanistic support for our conclusions regarding partial phenotype rescue. Full resolution of this technical limitation will require future plasmid redesign studies.

Reviewer #2 (Remarks to the Author):

1. I appreciate the authors' efforts to address my comments. All relevant questions appear to be answered sufficiently. I recommend this manuscript for acceptance.

Reply: Thanks very much for your constructive comments and hard work on the revision work that has been done.

2. Minor revisions: Figure 5. Please correct Gentamicin and tobramycin.

Reply: Thanks for the constructive advice. We have corrected the suggested information in the revised Figure 5.

COMMSBIO-24-5841 Comments:

In the manuscript titled "Systems-level exploitation of OxyR regulon unravels a novel antibacterial target in bacteria," the authors explore the role of the OxyR regulon in various cellular functions beyond its traditional regulatory role in oxidative stress responses. Through transcriptional and metabolic profiling, they discovered that OxyR positively regulates several genes involved in quorum sensing (QS) and energy metabolism. Additionally, OxyR negatively regulates amino acid transporters, suggesting that amino acid homeostasis represents a new aspect of the oxidative stress response in *Pseudomonas aeruginosa*. Under hypoxia conditions, *gltS* has been upregulated in the OxyR mutant. This observation led the authors to hypothesize the potential involvement of *gltS* during oxidative stress and this could be a probable target for antibiotic (bactericidal) drug potentiation.

The role of the OxyR regulon in oxidative stress responses and its regulatory functions in bacteria, including *Pseudomonas aeruginosa*, is already well-documented in existing literature. Previous studies have established OxyR's involvement in regulating genes related to oxidative stress, virulence, and quorum sensing (<https://doi.org/10.1128/iai.00837-09>, <https://doi.org/10.3390/antiox13060655>, <https://doi.org/10.1093/nar/gks017>). The new finding of this work is identification of *gltS* as a potential target for drug potentiation. But this part of the work is very weak and superficial, and substantial experiments need to be performed. In my view, this manuscript is preliminary and require a significant revision.

Comments:

1. Bacterial killing mechanism of bactericidal antibiotics is associated with ROS generation. Therefore, authors have tested if *gltS* mutant strain would become more sensitive to bactericidal antibiotics (Aminoglycoside antibiotic: tobramycin and gentamicin; beta-lactam antibiotic: ampicillin; Fluoroquinolone class: Ciprofloxacin). In the case of aminoglycosides and beta-lactams; drug potentiation has been observed. But no drug potentiation with ciprofloxacin. Fluoroquinolones are also bactericidal antibiotics. Surprisingly concentrations used to study bacterial killing are missing. Bactericidal experimental procedure also is missing.
2. Generally, PAO1 inherently harbours AmpC beta-lactamase that confers resistance to penicillins including ampicillin. I was wondering how authors were able to see bacterial killing effect with ampicillin.
3. Authors must perform experiments with Gram-negative active bacteriostatic antibiotics such as tetracycline class and compare their data with bactericidal drugs.
4. Authors have not performed experiments to check if the potentiation is indeed due to ROS overproduction in the presence and in the absence of antibiotics. In addition, radical scavengers such as dipyrityl should be included if the effect reverses.
5. Bacterial killing mechanism of bactericidal antibiotics is associated with ROS generation. Does this indicate that bactericidal antibiotics in general lead to impact the expression levels of *gltS*?
6. What is the fitness level of *gltS* mutant strain? Does it grow well similar to wildtype? What is the MIC of antibiotics used in the work against this mutant?
7. Further, authors should also verify the drug potentiation and other associated experiments with glutamine-depleted and glutamine-rich medium.
8. Provide details on how the *oxyR* and *gltS* mutants were constructed in supplementary information.

9. Please carefully check the manuscript for typographical errors. For example, “ESKAPE” in place of ESCAPE. Correct gentamicin throughout the manuscript. Gentamicin was first isolated from non-Streptomyces sp. therefore it is spelled as gentamicin not gentamycin.

Overall, the title “Identification of a Novel Antibacterial Target in Bacteria” and the conclusions drawn by the authors lack adequate supporting evidence. Therefore, this manuscript is incomplete, and I recommend for rejection.